# Metalation of a Hierarchical Self-Assembly Consisting of *π*-Stacked Cubes through Single-Crystal-to-Single-Crystal Transformation

**DOI:** 10.3390/molecules28134923

**Published:** 2023-06-22

**Authors:** Bin Wang, Zi-Ang Nan, Jin Liu, Zi-Xiu Lu, Wei Wang, Zhu Zhuo, Guo-Ling Li, You-Gui Huang

**Affiliations:** 1CAS Key Laboratory of Design and Assembly of Functional Nanostructures, and Fujian Provincial Key Laboratory of Nanomaterials, Fujian Institute of Research on the Structure of Matter, Chinese Academy of Sciences, Fuzhou 350002, China; wb@fjirsm.ac.cn (B.W.); xmliujin@fjirsm.ac.cn (J.L.); zxlu@mail.ustc.edu.cn (Z.-X.L.); wangwei@fjirsm.ac.cn (W.W.); zhuozhu@fjirsm.ac.cn (Z.Z.); 6469@cumt.edu.cn (G.-L.L.); 2Fujian Science & Technology Innovation Laboratory for Optoelectronic Information of China, Fuzhou 350108, China; 3Xiamen Key Laboratory of Rare Earth Photoelectric Functional Materials, Xiamen Institute of Rare Earth Materials, Haixi Institutes, Chinese Academy of Sciences, Xiamen 361021, China; 4University of Chinese Academy of Sciences, Beijing 100049, China

**Keywords:** self-assembly, *π*-stacked cube, metalation, single-crystal-to-single-crystal transformation, ion exchange

## Abstract

Single-crystal-to-single-crystal metalation of organic ligands represents a novel method to prepare metal–organic complexes, but remains challenging. Herein, a hierarchical self-assembly {(H_12_L_8_)·([N(C_2_H_5_)_4_]^+^)_3_·(ClO_4_^−^)_15_·(H_2_O)_32_} (**1**) (L = tris(2-benzimidazolylmethyl) amine) consisting of *π*-stacked cubes which are assembled from eight partially protonated L ligands is obtained. By soaking the crystals of compound **1** in the aqueous solution of Co(SCN)_2_, the ligands coordinate with Co^2+^ ions stoichiometrically and ClO_4_^−^ exchange with SCN^−^ via single-crystal-to-single-crystal transformation, leading to {([CoSCNL]^+^)_8_·([NC_8_H_20_]^+^)_3_·(SCN)_11_·(H_2_O)_13_} (**2**).

## 1. Introduction

Metal–organic complexes have been widely used in a lot of fields including luminescence [1,2,3,4,5,6,7,8], electrical conductivity [9,10,11,12,13], magnetism [14,15,16,17,18,19], catalysis [20,21,22,23,24,25], and so on. In these metal–organic-complex-relevant applications, the metal ions usually act as the functional centers. For example, metal ions can be the luminescent centers in luminescent materials [26], paramagnetic centers in magnetic materials [27], and the active centers in catalysts [28]. Therefore, the metal ions are usually critical to the functionalities of metal–organic complexes.

Metal–organic complexes are traditionally prepared homogeneously by dissolving inorganic metal salts and ligands in solution, and crystals of metal–organic complexes are harvested by crystallization. The crystalline morphology of a metal–organic complex prepared by this strategy is difficult to control. Metalation also occurs heterogeneously via single-crystal-to-single-crystal transformation [29,30,31,32,33]. In general, the metal–organic complexes prepared by this method preserve the parent morphology [30,31]. Therefore, the crystal structures of the produced metal–organic complexes can be easily predicted. Recently, such a post-synthetic metalation method has been used to mount metal active sites onto the open chelating sites of metal–organic frameworks (MOFs) to endow the framework with catalytic activity [32,33]. Specifically, MOFs with open chelating sites were firstly synthesized using linkers with hard carboxylate donors and a soft chelating moiety. Single-crystal-to-single-crystal metalation of the open chelating sites was then achieved under solvothermal conditions [32,33]. We envisage that this metalation method can be used to achieve hierarchical self-assembly bearing open chelating sites, which has not been reached.

Recently, the tripodal ligands tris(2-benzimidazolylmethyl) amine (L) and tris(2-naphthimidazolylmethyl) amine have attracted our attention because of their chelating sites formed by four nitrogen atoms [34,35,36,37]. The benzimidazolylmethyl arms enable L to self-assemble into *π*-stacked polyhedrons with open chelating sites through *π*···*π* interactions. These open chelating sites provide the possibility of single-crystal-to-single-crystal metalation. In this work, we report the single-crystal-to-single-crystal metalation of a hierarchical self-assembly {(H_12_L_8_)·([N(C_2_H_5_)_4_]^+^)_3_·(ClO_4_^−^)_15_·(H_2_O)_32_} (**1**) consisting of *π*-stacked cubes. Upon immersing the crystals of compound **1** into the aqueous solution of Co(SCN)_2_, Co^2+^ ions were mounted onto the chelating sites of L stoichiometrically and ClO_4_^−^ exchanged with SCN^−^ leading to the crystals of {([CoSCNL]^+^)_8_·([NC_8_H_20_]^+^)_3_·(SCN)_11_·(H_2_O)_13_} (**2**).

## 2. Results and Discussion

### 2.1. Structural Characterization of the Hierarchical Self-Assembly of Compound ***1***

The self-assembly of iron(II) perchlorate hydrate (Fe(ClO_4_)_2_·6H_2_O) with L in MeOH with traces of tetraethylammonium hydroxide ([N(C_2_H_5_)_4_]OH) affords yellow cubic crystals of {(H_12_L_8_)·([N(C_2_H_5_)_4_]^+^)_3_·(ClO_4_^−^)_15_·(H_2_O)_32_} (**1**) (Appendix A). The formula was determined by a combination of single-crystal X-ray crystallography (Appendix A) and TG analysis (Appendix A). Compound **1** crystallizes in the cubic space group *Fm*$\overset{-}{3}$*c*. The asymmetric unit contains 1/3 partially protonated L, 1/8 [N(C_2_H_5_)_4_]^+^, 5/8 ClO_4_^−^, and disordered H_2_O molecules. Surprisingly, the chelating site of L (Figure 1a) was not occupied by Fe^II^ during the self-assembly process which may be attributed to the fact that Fe^II^ is too soft to coordinate with the chelating site of L.

The crystal structure of compound **1** can be described as a hierarchical self-assembly consisting of *π*-stacked cubes, and the hierarchical structural complexity is best appreciated by describing it in a bottom-up fashion. As shown in Figure 1, eight partially protonated L ligands associate together through *π*···*π* interactions forming a homochiral *π*-stacked cube, and the cube [H_12_L_8_]^12+^ carries twelve protons to compensate the negative charges from ClO_4_^−^. Each partially protonated L locates on the vertices of the cube associating with its three neighbors (Figure 1b). The cubes with opposite handedness are connected by [N(C_2_H_5_)_4_]^+^ and ClO_4_^−^ forming a three-dimensional (3D) hierarchical self-assembly with [N(C_2_H_5_)_4_]^+^ and ClO_4_^−^ positioned at the interstitial regions between the cubes. Due to the simple cubic packing (SC) of the *π*-stacked cubes in lattice, there exists one-dimensional (1D) channels filled with disordered H_2_O molecules (Figure 1c).

### 2.2. Single-Crystal-to-Single-Crystal Metalation of Compound ***1***

Since we have obtained the supramolecular self-assembly with open chelating sites, we then try to verify the possibility of mounting metal ions onto the chelating sites via single-crystal-to-single-crystal transformation. Soaking in the aqueous solution of Co(SCN)_2_ (0.15 mol/L), the crystals gradually change from yellow to purple (Figure 2). The photographs showing the color change in the single crystal of compound **1** at different time intervals during the metalation process are shown in Appendix A. After being soaked for ten days, an intact crystal was picked out for structural determination. Single-crystal X-ray crystallography confirmed the successful metalation, and compound **1** transitioned to a similar hierarchical self-assembly {([CoSCNL]^+^)_8_·([NC_8_H_20_]^+^)_3_·(SCN)_11_·(H_2_O)_13_} (**2**), which consists of *π*-stacked cubes of {[CoSCNL]^+^}_8_. The formula was also determined by a combination of single-crystal X-ray crystallography (Appendix A) and TG analysis (Appendix A). A mixture of L (40.8 mg, 0.1 mmol), Co(SCN)_2_ (35.0 mg, 0.2 mmol), and tetraethylammonium hydroxide ([N(C_2_H_5_)_4_]OH) (0.06 mL) in 5 mL methanol was stirred at room temperature for 5 min. The resulting solution was placed in a beaker undisturbed at room temperature for volatile crystallization; however, compound **2** was not obtained.

The asymmetric unit of compound **2** contains 1/3 [CoSCNL]^+^, 1/8 [N(C_2_H_5_)_4_]^+^, 11/24 SCN^−^, and disordered H_2_O molecules. The *π*-stacked cube structure is preserved after metalation, and the size of the cube remains almost unchanged. The mounted Co^2+^ ion coordinates with four N atoms from the same L ligand and one SCN^−^ ion, exhibiting a trigonal-bipyramidal geometry (Figure 3a). All the coordinated SCN^−^ in the [CoSCNL]^+^ cations point inward to the center of the cube (Figure 3b). The crystal crystallizes in the same space group *Fm*$\overset{-}{3}$*c* before and after metalation, but the crystallographic axes significantly shrink from 39.96 to 38.66 Å (Appendix A, Figure 4). The unit cell contraction is cross-checked by the significant shift to higher angles of the powder X-ray diffraction (PXRD) peaks of compound **2** compared to those of compound **1**. For example, the peaks indexed to (220), (400), and (622) shift from 6.26°, 8.84°, and 15.28° in compound **1** to 6.44°, 9.04°, and 15.58° in compound **2**, respectively (Figure 5a). However, the counter anions ClO_4_^−^ at the interstitial regions between cubes in compound **1** were replaced by SCN^−^ in compound **2** (Figure 3c) which can also be revealed by the IR spectrum change (Figure 5b) from compound **1** to compound **2** [38,39]. The absorbances at 1020 and 620 cm^−1^ characteristic of ClO_4_^−^ become unobvious, and the absorbance at 2225 cm^−1^ characteristic of SCN^−^ appears. The side lengths of a cubic cage with the central N atom of the ligand L as the vertex are slightly expand after the single-crystal transformation (from 10.415 to 10.618 Å), respectively. We speculate that the anion exchange from tetrahedral ClO_4_^−^ to linear SCN^−^ is the major reason for the observed significant unit cell contraction (from 63,829 to 57,762 Å^3^) (Figure 4).

To clarify whether the metalation of the ligand Lis complete, a temperature-dependent magnetization study was performed on compound **2**. The *χ*_m_*T* value of 19.41 cm^3^ K mol^−1^ per *π*-stacked cube is higher than the spin-only value of eight isolated high-spin Co^II^ ions (15 cm^3^ mol^−1^ K) (Figure 5c) [40,41], suggesting all the L ligands are coordinated with Co^II^ ions. Upon cooling, the *χ*_m_*T* value keeps almost constant until 100 K, and then begins to decrease, reaching a value of 11.81 cm^3^ mol^−1^ K at 2 K, implying very weak antiferromagnetic coupling between Co^II^ ions. In the range of 2−300K, the magnetic susceptibility data follows the Curie−Weiss law, providing *θ* = −4.42 K and *C* = 19.47 cm^3^ K mol^−1^, confirming the dominant weak antiferromagnetic interaction between the Co^II^ ions. To provide further insight into the magnetism of compound **2**, the field-dependent magnetizations were measured. As shown in Figure 5d, the magnetization increases slowly with increasing field and reaches a value of 16.24 *Nβ* at 50 K Oe without obvious hysteresis, which is consistent with the weak antiferromagnetic couplings between Co^II^ ions [42,43].

## 3. Experimental

### 3.1. Materials and Physical Measurements

The ligand tris(2-benzimidazolylmethyl) amine (L) was synthesized according to the procedure reported in the literature [44], and all the other reagents were commercially obtained and used without further purification. Scanning electron microscopy (SEM; Hitachi SU1510, Chiyoda-ku, Tokyo, Japan) analysis was carried out on a Hitachi SU1510 SEM. Powder X-ray diffraction (PXRD; Miniflex 600, Akishima, Rigaku, Tokyo, Japan) patterns were performed on a Rigaku Miniflex 600 diffractometer with Cu-Kα radiation using flat plate geometry. Thermogravimetric analysis (TGA/DSC 1, Mettler Toledo, Zurich, Switzerland) was performed on a Mettler Toledo TGA/DSC 1 system with a heating rate of 10 K/min under an argon atmosphere. Fourier-transform infrared (FTIR; Nicolet iS 50, Thermo Fisher, Waltham, MA, USA) spectra were recorded in the range of 500–4000 cm^−1^ on a Thermo Nicolet is50 FT-IR spectrometer at room temperature. Magnetic measurements (MPMS-5S SQUID, Quantum Design, San Diego, CA, USA) were performed on a MPMS-5S SQUID magnetometer under an external field of 1000 Oe.

### 3.2. Synthesis of Compounds ***1*** and ***2***

Synthesis of {(H_12_L_8_)·([N(C_2_H_5_)_4_]^+^)_3_·(ClO_4_^−^)_15_·(H_2_O)_32_} (**1**): A mixture of L (40.8 mg, 0.1 mmol), Fe(ClO_4_)_2_·6H_2_O (76.4 mg, 0.3 mmol), and tetraethylammonium hydroxide ([N(C_2_H_5_)_4_]OH) (0.06 mL) in 5 mL methanol was stirred at room temperature for 5 min. The resulting solution was placed in a beaker undisturbed at room temperature. Yellow cubic crystals of compound **1** were obtained within seven days. Yield: 80% based on L ligand.

Synthesis of {([CoSCNL]^+^)_8_·([NC_8_H_20_]^+^)_3_·(SCN)_11_·(H_2_O)_13_} (**2**). Upon immersing the crystals of compound **1** in the aqueous of Co(SCN)_2_ (0.15 mol/L) for 10 days, the crystals of compound **2** were harvested as purple crystals.

### 3.3. Crystallography

Single-crystal X-ray data were harvested on a Bruker D8 Venture diffractometer with Mo-K_α_ radiation at 200 K. Structures were solved using a direct method and refined by the full-matrix least-squares technique on F^2^ with the SHELXTL 2014 program [45]. All the H atoms were geometrically generated and refined using a riding model. The X-ray crystallographic coordinates for structures reported in this article have been deposited at the Cambridge Crystallographic Data Centre (CCDC) under deposition numbers 2266971–2266972. These data can be obtained free of charge from The Cambridge Crystallographic Data Centre via www.ccdc.cam.ac.uk/data_request/cif (accessed on 2 June 2023). Detail crystallographic data are listed in Appendix A.

## 4. Conclusions

In conclusion, we have synthesized a hierarchical self-assembly consisting of *π*-stacked cubes with open chelating sites. Learning from the single-crystal-to-single-crystal metalation of MOFs, metal ions were successfully mounted onto the chelating sites via single-crystal-to-single-crystal transformation. This strategy may be used for metalation of other supramolecular frameworks with open coordination sites, thus providing a new method of synthesizing metal–organic complexes.

## Figures and Tables

**Figure 1 molecules-28-04923-f001:**
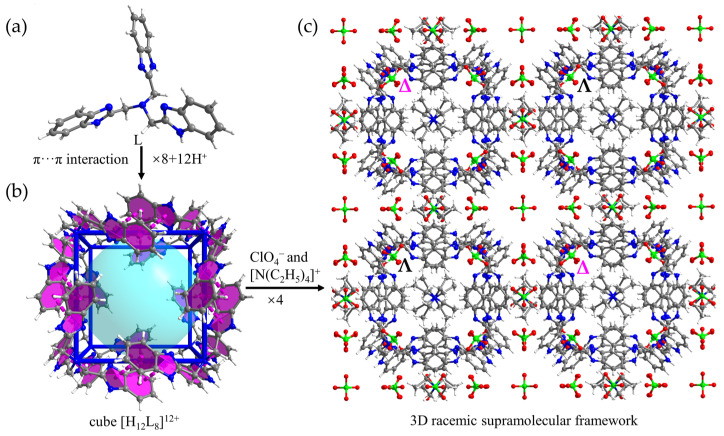
Hierarchical structure of the compound **1**. (**a**) The structure of the isolated ligand L. (**b**) The cubic cage [H_12_L_8_]^12+^ formed by eight partially protonated ligands L (the *π*···*π* interactions are indicated by purple dotted lines). (**c**) The 3D hierarchical self-assembly with [N(C_2_H_5_)_4_]^+^ and ClO_4_^−^ positioned at the interstitial regions between cubes. (inset: adjacent cubic cages with opposite chirality are denoted by ∆ and Λ, respectively).

**Figure 2 molecules-28-04923-f002:**
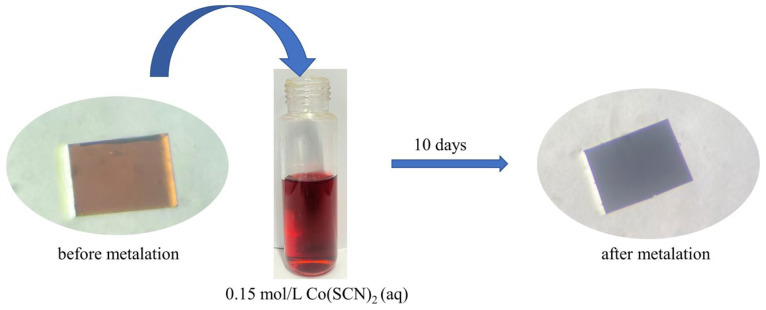
The photographs showing the single-crystal-to-single-crystal transformation from compound **1** to compound **2**.

**Figure 3 molecules-28-04923-f003:**
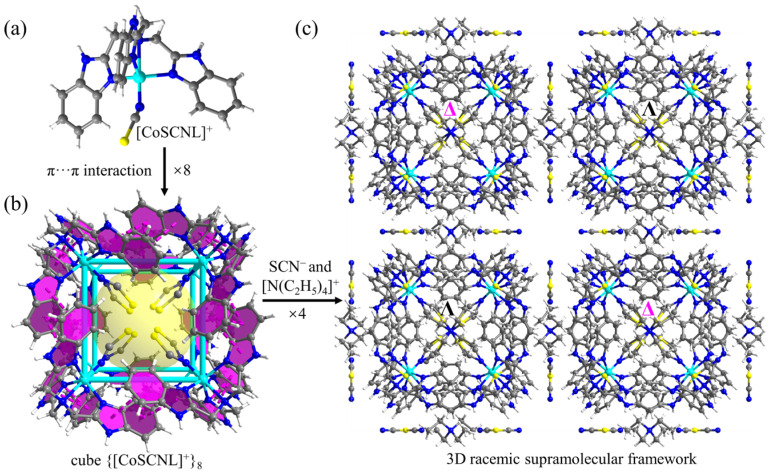
Structure of compound **2**. (**a**) Structure of [CoSCNL]^+^. (**b**) The *π*-stacked cubes of {[CoSCNL]^+^}_8_ in compound **2** (the *π*···*π* interactions are indicated by purple dotted lines). (**c**) The 3D hierarchical structure of compound **2** with [N(C_2_H_5_)_4_]^+^ and SCN^−^ positioned at the interstitial regions between the cubes. (inset: adjacent cubic cages with opposite chirality are denoted by ∆ and Λ, respectively).

**Figure 4 molecules-28-04923-f004:**
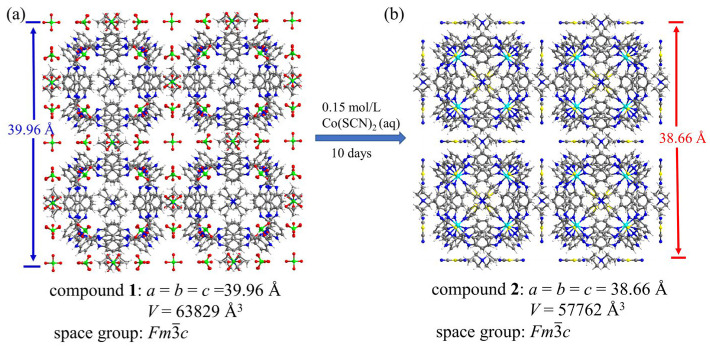
Schematic diagram of unit cell and unit cell parameters changes of compound **1** and compound **2** before and after metalation through single-crystal-to-single-crystal transformation. (**a**) The unit cell and unit cell parameters of compound **1**. (**b**) The unit cell and unit cell parameters of compound **2**.

**Figure 5 molecules-28-04923-f005:**
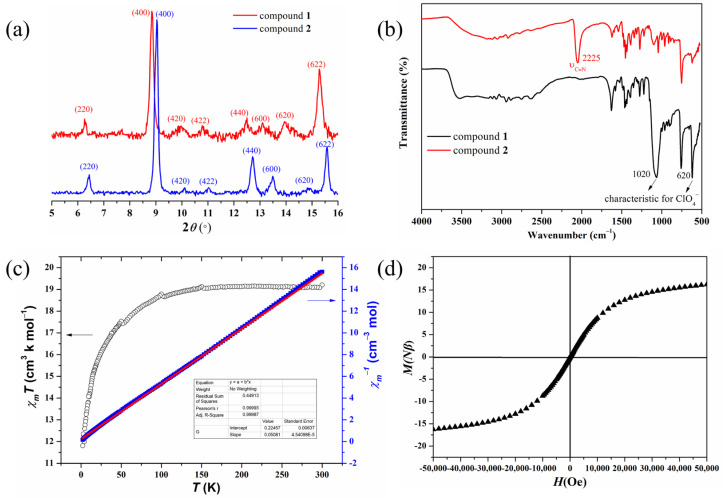
(**a**) PXRD patterns of compounds **1** and **2**. (**b**) IR spectra of compounds **1** and **2**. (**c**) The plot of *χ*_m_*T* – *T* and *χ*_m_^−1^ – *T* of compound **2**. (**d**) The plot of *M* − *H* of compound **2**.

## Data Availability

All data related to this study are presented in this publication.

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
