# Peer review of "Metalation of a Hierarchical Self-Assembly Consisting of π-Stacked Cubes through Single-Crystal-to-Single-Crystal Transformation"

_molecules, 2023, doi:10.3390/molecules28134923_

Round 1
Reviewer 1 Report
The paper of You-Gui Huang and co-authors is an interesting fundamental work on synthesis organic polymer by self-assembly method with further transformation of its crystals with another compound containing cobalt thiocyanate. The authors were lucky to isolate crystals 1, apparently, at first they wanted to obtain an iron organometallic framework. Further, the authors managed to modify the crystals of compound 1 by soaking the crystals in a solution of cobalt thiocyanate. And this was confirmed by single-crystal X-ray diffraction analysis. The article is written very well, neatly, I have no major remarks.
Nevertheless I have questions to the bibliography. I did not analyze all the sources, but what I managed to find.
Lines 36-37: “Metalation also occurs heterogeneously via single-crystal-to-single-crystal transformation [29‒39]” but
Ref 31: in this article, the authors study the transformation of a single crystal into a single crystal of a cobalt compound [Co(H2O)4]Cl2 during solvation or desolvation figure 4 (not metalation). At the same time, there is no information on metalation with the transformation of crystals. Only information about the keeping of crystallinity, which is not proof of the transformation of some crystals into others.
Ref 32: In this work, there is also no information on the transformation of single crystals into a single crystal. There is powder diffraction, which indicates the crystallinity of the product, but does not prove the transformation of crystals into others. “The accessibility of the chelating bpy units within the framework of 1 was confirmed by soaking the solid in acetonitrile solutions of PdCl2 and Cu(BF4)2 to afford 1 · xPdCl2 (x ) 0.08, 0.83) and 1 · 0.97Cu(BF4)2, as confirmed by elemental analyses. Thermogravi[1]metric analysis showed these metal-loaded samples to exhibit weight losses at 125 °C of 28%, 15%, and 25%, respectively, with no additional loss occurring up to 350 °C (see Supporting Information,Figure S8). As evidenced by powder X-ray diffraction, NMR spectroscopy, and infrared spectroscopy, the underlying framework structure is maintained upon metal complexation.”
Ref 34: This work is completely irrelevant in the list of citations on the transformation of a single crystal into a single crystal. So as it talks about the reaction of porphyrin with zinc nitrate, which takes place in solution. “By metalation of the porphyrin tecton,29, 30 the morphology of the nanoparticles can be controlled. For example, by addition of Zn2+ (through Zn(NO3)2) in the self-assembly solution, Zn inserts into the core of H2TPyP through incorporation of Zn ions with the core pyrroles forming ZnTPyP. Subsequent self-assembly of ZnTPyP was conducted through axial ligation between center Zn and pyridyl groups. In a typical procedure for metalation of H2TPyP to form ZnTPyP, a stock solution was prepared with 0.01 M H2TPyP dissolved in 0.2 M HCl and stirred for 24 hours. Zn(NO3)2 was then added with a molar ratio to H2TP”
Ref 35: The paper also discuss only the crystallinity of the obtained samples. The authors do not claim the transformation of single crystals into a single crystal. “Powder X-ray diffraction (PXRD) indicated that the crystallinity of 1 was maintained in both 1·Rh and 1·Ru after the metalation reactions”
Ref 36: This work is completely irrelevant in the list of citations on the transformation of a single crystal into a single crystal. It talks about the use of organometallic frameworks in catalysis and the keeping of crystallinity after the catalysis process. Perhaps the authors wanted to cite another work, such as this P. V. Dau, M. Kim and S. M. Cohen, Chem. Sci., 2013, 601–605
Ref 37: In this work, there is also no information on the transformation of single crystals into a single crystal. There is powder diffraction, which indicates the crystallinity of the product, but does not prove the transformation of crystals into others. “The crystallinity of bpy-UiO was maintained in both bpy-UiO-Ir and bpy-UiO-Pd as shown by their PXRD patterns”
In any case, these remarks refer only to publication ethics and do not affect the scientific value of this work. The presented manuscript is a very interesting work and I certainly recommend accepting it for publication in the journal Molecules. This work will definitely attract a lot of attention from researchers and I hope increase citations for the Molecules.
Author Response
Response: Thank you for your suggestion! The literatures of 31, 32 and 34−37 are removed in our revised manuscript.
Reviewer 2 Report
The authors presented the metalation of a self-assembly of N-donor ligands in a single-crystal-to-single-crystal transformation. This interesting strategy allowed them to maintain the overall cubic crystal structure after coordination of Co(II) ions. The literature has been correctly addressed, the magnetic susceptibility and field-dependent magnetization measurements confirmed the presence of almost isolated CoII ions. More importantly, the crystal structures could be resolved despite a low crystallinity and large volume of the unit cells. Such problems naturally result in very high R1/wR2 values, and to a goodness of fit parameter far from unity (>2.2, surprisingly leading to an alert of level C only), probably because of a low reflections/parameters ratio and high number of restraints. However, the complexity of such systems can justify the presence of level A alerts. For convenience, the following information should be provided in the crystallographic data: Rint, number of parameters and restraints. The authors should also correct the sentence “FeII is too soft to coordinate with the hard FeII”.
Overall, I recommend the publication of the manuscript in Molecules.
Author Response
Response: Thank you for your suggestion! The information of “Rint, number of parameters and restraints” are added in the crystallographic data of our revised supplementary information. These are presented as “Rint is 0.0407, Rsigma is 0.0167, Data/restraints/parameters are 1540/167/253 for compound 1; Rint is 0.0400, Rsigma is 0.0051, Data/restraints/parameters are 2231/42/192 for compound 2.” in our revised supplementary information.
We have corrected this written error. These are presented as “FeII is too soft to coordinate with the chelating site of L.” in our revised manuscript.
Reviewer 3 Report
This paper reports the single-crystal-to-single-crystal transformation from cubic cage (1) to Co-functionalized metal-organic cage (2) by the introduction of Co(II) ions in vacant coordination sites of L in cubic cage. The structures of the cubic cage (1) and Co-cage (2) were characterized by single crystal X-ray analyses. Further characterization of the Co-complex was made by powder X-ray analysis IR spectroscopy. The authors also investigated the magnetic property of the crystals of 2. The followings are several comments on this manuscript to improve the value of this manuscript.
1. It is better for the authors to discuss why the crystallographic axes were shrink by metalation.
2. The authors reported that it took 10 days for making Co-cage by metalation of the cubic cage 1. I wonder if the authors can monitor the metallation process of 1 by some experimental techniques.
3. The description of Figures 1 and 2 may mislead the readers as if the crystallization process is shown in these figures.
4. The authors confirmed that the complexation of L and Fe(II) ion did not proceed. How about the complexation of L with Co(II) ion? Is the 1:1 complex between them produced as shown in Figure 3a? If possible, how about the crystallization of the [CoSCNL]+ complex? Is the same CO-cubic cage (2) produced from the crystallization of [CoSCNL]+?
Some typos were found in the manuscript.
